# Exploring Augmentation-Driven Invariances for Graph Self-supervised Learning in Spatial Omics

**Lovro Rabuzin**\* **Michel Tarnow**\* **Valentina Boeva**
Department of Computer Science, ETH Zurich
{lrabuzin, mtarnow, vboeva}@ethz.ch

## Abstract

Spatial omics technologies provide rich insights into biological processes by jointly capturing molecular profiles and the spatial organization of cells. The resulting high-dimensional data can be naturally represented as graphs, where Graph Neural Networks (GNNs) offer an effective framework to model interactions in the tissue. Self-supervised pretraining methods such as Bootstrapped Graph Latents (BGRL) and GRACE leverage graph augmentations to build invariances without costly labels. Yet, the design of augmentation strategies remains underexplored, particularly in the context of spatial omics. In this work, we systematically investigate how different graph augmentations affect embedding quality and downstream performance in spatial omics. We evaluate a suite of existing and novel augmentations, including transformations tailored to biological variation, across two representative tasks: unsupervised domain identification in healthy tissue and supervised phenotype prediction in cancer tissue. Our results show that carefully chosen augmentations substantially improve performance, whereas poorly aligned or overly complex augmentations may fail to help or even degrade performance. These findings highlight the central role of augmentation design in enforcing meaningful invariances for graph contrastive pretraining in spatial omics.

## 1 Introduction

Spatial omics technologies measure molecular profiles, such as RNA or protein expression, while preserving the spatial context of cells in their natural environment. This modality provides a more comprehensive view of cellular behavior, biological processes, disease mechanisms, and therapeutic responses compared to non-spatial single-cell methods [1]. Spatial transcriptomics platforms like multiplexed error-robust fluorescence in situ hybridization (MERFISH) [2], spatially-resolved transcript amplicon readout mapping (STARmap) [3], Xenium [4], and barcode in situ targeted sequencing (BaristaSeq) [5] use microscopy or in situ sequencing to generate spatial maps of RNA expression. Complementary proteomics methods such as imaging mass cytometry (IMC) [6] and co-detection by indexing (CODEX) [7] measure protein abundances with spatial resolution.

The complex and high-dimensional data produced by these technologies can be naturally represented as graphs, where nodes correspond to cells and edges encode spatial proximity or molecular similarity [8]. To exploit all available information from spatial omics data, graph-based methods like graph neural networks (GNNs) often exhibit superior characteristics compared to traditional analysis methods not taking spatial dependencies in the data into account [9, 10]. GNNs are well-suited to analyze spatial omics data, as they explicitly model relationships between cells through graph structures using a message-passing mechanism [9].

---

\*Equal contribution. Code available at `https://github.com/BoevaLab/spatial-augmentations`

Pretraining enables GNNs to learn generalizable patterns from data before fine-tuning them for specific tasks. Self-supervised or unsupervised pretraining methods are especially valuable in biological contexts, where labeled data can be scarce and expensive [11]. Moreover, these approaches can introduce inductive biases, for instance via graph augmentations, that help models prioritize biologically relevant features and improve robustness [12].

A central principle of contrastive self-supervision is that it enforces invariance to augmentations: two different views of the same input are trained to have similar embeddings. Early works such as SimCLR [13] and its follow-ups demonstrated the effectiveness of this paradigm in computer vision by treating augmented images as positives and enforcing representation consistency. Subsequent methods like BYOL [14] removed the need for explicit negatives while still relying on augmented views to build invariances. The choice of augmentations defines the invariances learned by the model, in line with the *InfoMin principle* that views should remove nuisance factors but preserve task-relevant information [15]. Recent work has shown that in graph domains, augmentations explicitly inject desired invariances, such as robustness to node/edge perturbations or feature corruption [12, 16].

Several pretraining frameworks have operationalized these ideas in the graph setting. Deep Graph Contrastive Representation Learning (GRACE) [16] builds invariance to structural and feature perturbations via an InfoNCE-based contrastive loss. Bootstrapped Graph Latents (BGRL) [17] achieves similar invariances without negative samples, relying instead on online–target encoder consistency. Both methods demonstrate that invariances induced by carefully chosen augmentations significantly enhance representation quality and downstream performance. While recent benchmarks such as scSSL-Bench [18] have evaluated self-supervised learning in a biological context across diverse single-cell omics modalities, spatial omics remains underexplored. Most existing applications of GNNs to spatial omics adopt generic augmentations from other domains or do not leverage augmentation at all [8, 19–21].

This work explores how different graph augmentation strategies affect the quality of node and graph embeddings in spatial omics data. We investigate both existing graph augmentations, as well as newly designed augmentations that encode biologically meaningful inductive biases. Their effectiveness is evaluated on two representative downstream tasks in spatial omics: unsupervised domain identification on healthy tissue and supervised phenotype prediction in cancer samples. These tasks differ not only in supervision regime but also in biological complexity: domain identification on healthy tissue emphasizes stable spatial compartments, while phenotype prediction on cancer tissue must contend with tissue heterogeneity and noisy clinical labels [22–24]. To our knowledge, this is the first systematic investigation of graph augmentations in spatial omics, introducing novel biologically motivated transformations that explicitly encode inductive biases such as cellular plasticity and spatial measurement variability.

## 2 Methods

Figure 1 provides a schematic overview of our study design. Graph augmentations (baseline and advanced) are applied to input graphs, models are pretrained using BGRL and GRACE, and evaluated on two downstream tasks: domain identification and phenotype prediction. The following subsections describe each component in detail.

**Notations** A graph is denoted by $\mathbf{G} = (\mathbf{X}, \mathbf{A})$, where $\mathbf{X} \in \mathbb{R}^{N \times F}$ is the node feature matrix with $N$ nodes and $F$ features per node, and $\mathbf{A} \in \mathbb{R}^{N \times N}$ is the binary adjacency matrix. Graph augmentations generate a new view $\tilde{\mathbf{G}} = (\tilde{\mathbf{X}}, \tilde{\mathbf{A}})$ by modifying $\mathbf{X}$, $\mathbf{A}$, or both. Node positions are encoded in a spatial matrix $\mathbf{P} \in \mathbb{R}^{N \times d}$, with $d = 2$, yielding $\tilde{\mathbf{P}}$ after augmentation. The neighborhood of node $i$, denoted $\mathcal{N}(i)$, is defined as the set of nodes directly connected to $i$ in $\mathbf{A}$.

### 2.1 Baseline augmentations

Two baseline augmentations were used: **DropFeatures** and **DropEdges**. Models trained with these augmentations served as baselines for performance comparisons. All augmentation hyperparameters were tuned on validation sets over fixed ranges.

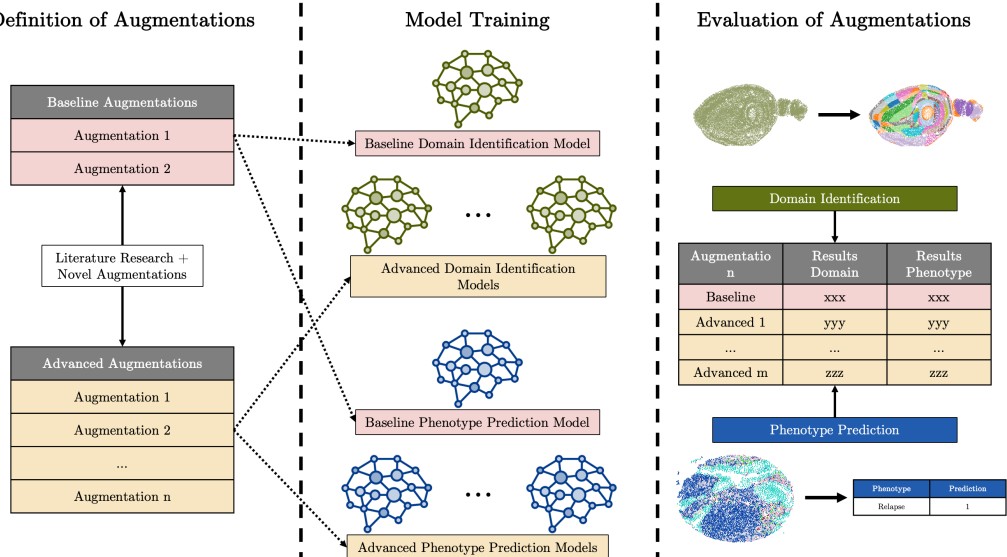

Figure 1: **Overview of the evaluation procedure.** Graph augmentations (baseline and advanced) are applied to input graphs. Models are pretrained using the BGRL and GRACE frameworks, then evaluated on two downstream tasks: domain identification and phenotype prediction.

**DropFeatures** randomly masks features by setting entries in $\mathbf{X}$ to zero with probability $p$, resulting in $\tilde{\mathbf{X}}$ while keeping $\mathbf{A}$ unchanged. If $\mathbf{X}$ contains a cell type feature, it is masked by setting entries to the numeric code of the "unassigned" type.

**DropEdges** randomly removes edges from $\mathbf{A}$ with probability $p$ (Bernoulli sampling), resulting in $\tilde{\mathbf{A}}$ while keeping $\mathbf{X}$ unchanged.

## 2.2 Advanced augmentations

Advanced augmentations include both published and novel methods. These were tested individually and in combination to assess their effect on downstream tasks relative to baseline augmentations.

**DropImportance** masks node features and removes edges based on importance scores. Inspired by prior work [25, 26], it is controlled by dropout rate $\mu$ and threshold $\lambda_p$. Node importance is derived from log-degree centrality:

$$I_i^{(n)} = \frac{\log(1 + \deg_i) - \bar{d}}{\max_j \log(1 + \deg_j) - \bar{d}}, \tag{1}$$

where $\deg_i$ is the degree of node $i$ and $\bar{d}$ is the mean log-degree. The node feature drop probability is

$$p_i = \min((1 - I_i^{(n)}) \cdot \mu, \lambda_p). \tag{2}$$

Edges are ranked by the mean importance of their endpoints,

$$I_{ij}^{(e)} = \tfrac{1}{2}(I_i^{(n)} + I_j^{(n)}), \tag{3}$$

normalized, and dropped with probability

$$p_{ij} = \min((1 - I_{ij}^{(e)}) \cdot \mu, \lambda_p). \tag{4}$$

This encourages invariance to the removal of less informative node features and edges.

**SpatialNoise** adds Gaussian noise to spatial positions:

$$\tilde{\mathbf{p}}_i = \mathbf{p}_i + \boldsymbol{\epsilon}_i, \quad \boldsymbol{\epsilon}_i \sim \mathcal{N}(\mathbf{0}, \sigma^2 \mathbf{I}). \tag{5}$$

This models experimental imprecision in cell localization and enforces invariance to small spatial perturbations. This augmentation is applicable only to tasks with spatial coordinates (domain identification).

**FeatureNoise** adds Gaussian noise to node features:

$$\tilde{\mathbf{x}}_i = \mathbf{x}_i + \boldsymbol{\epsilon}_i, \quad \boldsymbol{\epsilon}_i \sim \mathcal{N}(\mathbf{0}, \sigma^2 \mathbf{I}). \tag{6}$$

This simulates variability in molecular readouts and enforces robustness to minor fluctuations in expression.

**SmoothFeatures** applies a convex combination of each node's features with the mean of its neighbors:

$$\tilde{\mathbf{x}}_i = (1 - \alpha)\mathbf{x}_i + \alpha \cdot \frac{1}{|\mathcal{N}(i)|} \sum_{j \in \mathcal{N}(i)} \mathbf{x}_j, \tag{7}$$

where $\alpha \in [0, 1]$ controls the smoothing strength. This simulates transcript leakage [27] and enforces invariance to local feature diffusion. This augmentation is used only for domain identification.

**PhenotypeShift** randomly mutates discrete cell-type features $c_i$ according to a transition map $\mathcal{M}$:

$$\tilde{c}_i = \begin{cases} c_i & \text{with probability } 1 - p, \\ \text{sample}(\mathcal{M}[c_i]) & \text{with probability } p, \end{cases} \tag{8}$$

where $\mathcal{M}[c_i] \subseteq \mathcal{C}$ contains plausible phenotype alternatives. This models both plasticity (cell-type switching) and misclassification noise, training robustness to annotation uncertainty. This augmentation is used only for phenotype prediction. Details of $\mathcal{M}$ are dataset-specific.

## 2.3 The task of domain identification

The first task employed to evaluate augmentations is unsupervised *Domain Identification*. The objective is to detect and segment spatially coherent regions within healthy tissue based on molecular data (e.g., gene expression) and spatial data (e.g., spatial relationships). These regions, or domains, ideally reflect biologically relevant structures such as tissue compartments or functional zones.

### 2.3.1 Data

We used three spatial transcriptomics datasets with expert domain annotations, obtained via the benchmarking study of Schaub *et al.* (2025) [28]. Dataset details are summarized in Table 1.

Table 1: **Datasets used for the domain identification task.**

| Dataset | Technology | Samples | Cells |
|---------|------------|---------|--------|
| 1 | MERFISH | 5 | 28,317 |
| 2 | STARmap | 4 | 4,397 |
| 3 | BaristaSeq | 3 | 5,257 |

Dataset 1 profiles 5 mouse brain samples via MERFISH [29]. Dataset 2 contains STARmap data from mouse cortex [3], with expert annotations by Li and Zhou (2022) [30]. Dataset 3 comprises BaristaSeq samples of mouse cortex tissue [31]. All datasets are publicly available [32].

### 2.3.2 Pipeline and model

An overview of the domain identification pipeline is shown in Figure 2. Each sample is preprocessed into a graph, passed through a GCN encoder pretrained with BGRL or GRACE with spatial regularization, and clustered into domains using the Leiden algorithm [33].

**Data preprocessing and graph construction**  Each sample is first preprocessed using a sequence of filtering and normalization steps. Genes are filtered based on the number of cells they are detected in, and cells are filtered based on the number of genes they express. Cells lacking domain annotations are removed. Raw gene expression values are then normalized to a target sum of $10^5$ counts per cell, log-transformed, and scaled to unit variance and zero mean. Principal Component Analysis (PCA) is subsequently applied to the processed expression matrix.

Following preprocessing, one spatial omics graph is constructed per sample. Each cell is represented as a node, with the top 50 principal components (PCs) serving as node features. Nodes are connected to their $k$ nearest neighbors in Euclidean space, with edge weights uniformly set to 1. The number of neighbors $k$ is optimized during hyperparameter search.

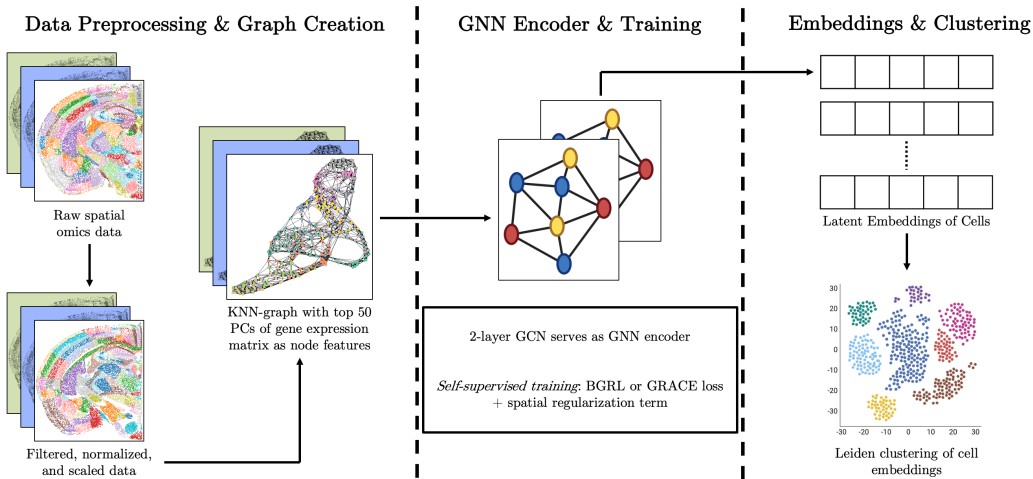

Figure 2: **Domain identification pipeline.** Each sample is preprocessed into a graph, passed through a GCN encoder pretrained with BGRL or GRACE and spatial regularization, and clustered into domains via the Leiden clustering method.

**Model and training**    A two-layer Graph Convolutional Network (GCN) is used to compute node embeddings for each sample-specific graph. The network is trained in a self-supervised manner using both the BGRL and GRACE frameworks.

To encourage spatial coherence in the learned representations, a spatial regularization term is added. It penalizes high similarity in the embedding space for nodes that are spatially distant. This discourages long-range spurious similarities. The resulting overall loss function is defined as:

$$\mathcal{L}_{\text{total}} = \mathcal{L}_{\text{SSL}} + \gamma_{\text{spatial}} \cdot \frac{1}{N^2} \sum_{i,j} \mathbf{D}_{i,j}^{(s)} \cdot (1 - \mathbf{D}_{i,j}^{(z)}), \tag{9}$$

where $\mathbf{D}_{i,j}^{(s)}$ denotes the normalized Euclidean distance between cells $i$ and $j$, and $\mathbf{D}_{i,j}^{(z)}$ denotes the normalized distance between their embeddings in the latent space. The regularization strength is controlled by the hyperparameter $\gamma_{\text{spatial}}$.

Self-supervised training is conducted across all data samples. For model selection and evaluation, the dataset is split into 40% validation and 60% test samples. Hyperparameters such as learning rate and spatial regularization strength are optimized using a validation-based hyperparameter search. The model is trained using the Adam optimizer with a cosine annealing learning rate scheduler.

**Clustering**    To obtain the final domain assignments for each node, the learned node embeddings are clustered using the Leiden algorithm [33]. The resolution parameter of the Leiden clustering is dynamically adjusted to match the number of ground truth domains in each sample. The resulting predicted domain labels are then evaluated against the ground truth annotations using clustering quality metrics. Metrics are calculated per sample and averaged across the validation or test set to report the overall performance. All reported means and standard deviations are computed over 5 independent runs with different random seeds.

## 2.4   The task of phenotype prediction

The second task used to evaluate augmentations is supervised *Phenotype Prediction* in human non-small cell lung cancer (NSCLC) tissue. The objective is to predict biological or clinical phenotypes directly from spatially resolved molecular data. Here, we predict cancer relapse after treatment. A detailed description of the data, model, and evaluation strategy for this task is provided below.

### 2.4.1 Data

The data used for phenotype prediction consists of one non-small cell lung cancer (NSCLC) spatial proteomics dataset obtained by imaging mass cytometry [34]. Marker expression was quantified with 45 metal-labeled antibodies in 1071 patients with at least 15 years follow-up, resulting in 1868 cancer samples. Each sample includes clinical annotations, for instance smoking status, cancer stage, relapse, clinical outcome, or cancer subtype. The raw data can be downloaded from the resource provided by Cords *et al.* (2024) [34].

### 2.4.2 Pipeline and model

The phenotype prediction pipeline is based on SPACE-GM [8]. An overview of the pipeline is shown in Figure 3.

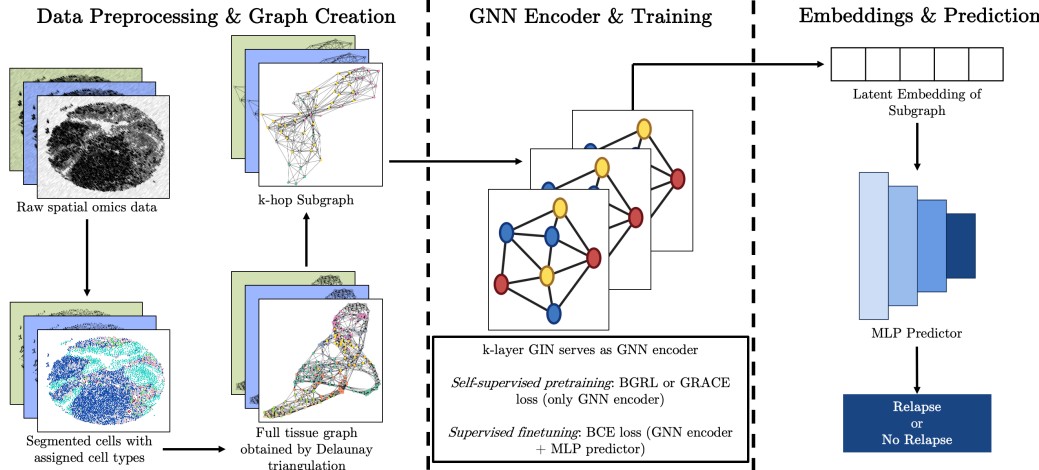

Figure 3: **Phenotype prediction pipeline.** Tissue graphs are built from omics data, subgraphs extracted, passed through a GIN encoder pretrained with BGRL or GRACE, and classified with an MLP.

**Data preprocessing and graph creation**    Graphs were constructed from segmented cells using Delaunay triangulation. Node features included cell type (integer-encoded) and cell size. Edge features consist of a binary "near/distant" category based on centroid-to-centroid distance, using a threshold of 20 $\mu$m, reflecting the typical size of human cells. From each tissue graph, $h$-hop subgraphs were extracted ($h = 3$ by default).

**Model and training**    An $L$-layer Graph Isomorphism Network (GIN) with edge-feature extension [8] was used, where messages are defined as

$$m_{vu}^{(\ell)} = h_u^{(\ell-1)} + e_{vu}^{(\ell)}, \tag{10}$$

with $e_{vu}^{(\ell)}$ mapped via an embedding lookup. Subgraph embeddings were obtained by max-pooling over final-layer node embeddings. The encoder was pretrained with BGRL and GRACE. For classification, a 3-layer MLP was added and jointly fine-tuned with a weighted BCE loss.

All experiments were run on a cluster with NVIDIA RTX 4090 GPUs (24GB) and 16-core CPUs, with up to 96GB RAM per job. Each training and evaluation run had a runtime of up to 4 hours.

**Splits and optimization**    Pretraining used all samples without labels. For supervised fine-tuning, 1492 samples were used for training and 376 for evaluation, with evaluation split 50% validation / 50% test. Splits were stratified by relapse, and all samples from a patient were assigned to the same fold. The performance was evaluated against the ground truth patient labels using standard classification quality metrics. Hyperparameters were tuned on the validation set. All reported means and standard deviations are computed on the test set over 5 independent runs with different random seeds.

## 2.5 Representation analysis

We assessed how augmentations influence the geometry of the learned embedding space by comparing encoder representations across trained BGRL models. For each augmentation configuration, we extracted node embeddings from the encoder on the evaluation set, resulting in an $N$ by $d$ matrix of representations.

To compare these embeddings, we used linear Centered Kernel Alignment (CKA)[35]. CKA evaluates how similarly two representation spaces preserve pairwise relationships between samples by comparing their centered Gram matrices. It is invariant to rotations and uniform scaling, which makes it suitable for assessing changes in geometry induced by different augmentations. For each task, we computed linear CKA between the embedding matrices produced by all pairs of augmentation configurations. The resulting pairwise similarity matrices show how closely the different learned embedding spaces resemble each other.

# 3 Results

## 3.1 Unsupervised domain identification in healthy mouse brain tissue

We first evaluated the effect of augmentations on domain identification in healthy mouse brain tissue. Baseline models were trained with *DropFeatures* and *DropEdges*, and compared against models with advanced augmentations or their combinations. The *Noise* augmentation denotes the joint application of *SpatialNoise* and *FeatureNoise*. Performance was measured using normalized mutual information (NMI), homogeneity (HOM), and completeness (COM).

Results with BGRL and GRACE are shown in Tables 2 and 3. Under BGRL, *DropImportance* improved NMI from 0.61 (baseline) to 0.66, with the next-best performance achieved by combining all augmentations (0.65). Under GRACE, *DropImportance* again achieved the best result (0.66 compared to 0.65 baseline), and was the only augmentation regime that substantially improved over the baseline. Across both frameworks, *DropImportance* provided the most consistent gains. With BGRL, nearly all augmentation regimes improved upon the baseline, whereas in GRACE the gains were smaller because the baseline was already comparatively strong.

Table 2: **Performance on domain identification task using BGRL.** Clustering performance on healthy mouse brain tissue using different augmentation strategies. Reported as mean $\pm$ standard deviation across 5 random seeds. The best and second-best results by mean are highlighted.

| Augmentations | NMI | HOM | COM |
|---|---|---|---|
| Baseline | $0.6145 \pm 0.0195$ | $0.6188 \pm 0.0234$ | $0.6121 \pm 0.0175$ |
| Baseline + Noise | $0.6488 \pm 0.0083$ | $0.6419 \pm 0.0093$ | $0.6576 \pm 0.0074$ |
| DropImportance | $0.6585 \pm 0.0033$ | $0.6552 \pm 0.0065$ | $0.6635 \pm 0.0008$ |
| DropImportance + Noise | $0.6488 \pm 0.0166$ | $0.6507 \pm 0.0135$ | $0.6498 \pm 0.0217$ |
| SmoothFeatures | $0.6497 \pm 0.0065$ | $0.6465 \pm 0.0097$ | $0.6538 \pm 0.0061$ |
| DropImp. + Noise + SmoothFeat. | $0.6540 \pm 0.0104$ | $0.6507 \pm 0.0110$ | $0.6579 \pm 0.0103$ |

Table 3: **Performance on domain identification task using GRACE.** Clustering performance on healthy mouse brain tissue using different augmentation strategies. Reported as mean $\pm$ standard deviation across 5 random seeds. The best and second-best results by mean are highlighted.

| Augmentations | NMI | HOM | COM |
|---|---|---|---|
| Baseline | $0.6470 \pm 0.0081$ | $0.6475 \pm 0.0081$ | $0.6484 \pm 0.0110$ |
| Baseline + Noise | $0.6405 \pm 0.0221$ | $0.6390 \pm 0.0157$ | $0.6438 \pm 0.0271$ |
| DropImportance | $0.6639 \pm 0.0056$ | $0.6569 \pm 0.0082$ | $0.6726 \pm 0.0046$ |
| DropImportance + Noise | $0.6477 \pm 0.0125$ | $0.6409 \pm 0.0120$ | $0.6557 \pm 0.0127$ |
| SmoothFeatures | $0.6460 \pm 0.0100$ | $0.6423 \pm 0.0115$ | $0.6509 \pm 0.0085$ |
| DropImp. + Noise + SmoothFeat. | $0.6412 \pm 0.0058$ | $0.6336 \pm 0.0050$ | $0.6502 \pm 0.0080$ |

Qualitative results of the models trained using BGRL are shown in Figure 4 for a representative MERFISH sample. Different augmentation strategies produce visibly different domain segmentations, broadly consistent with the quantitative metrics.

Overall, domain identification highlights how augmentations can improve unsupervised discovery of spatial structure in healthy tissue. To complement this, we next evaluate phenotype prediction, a supervised task in noisy and heterogeneous human cancer tissue. Together, these tasks represent distinct regimes, unsupervised and supervised, as well as healthy and cancerous tissue, which help reveal when particular augmentations are most beneficial.

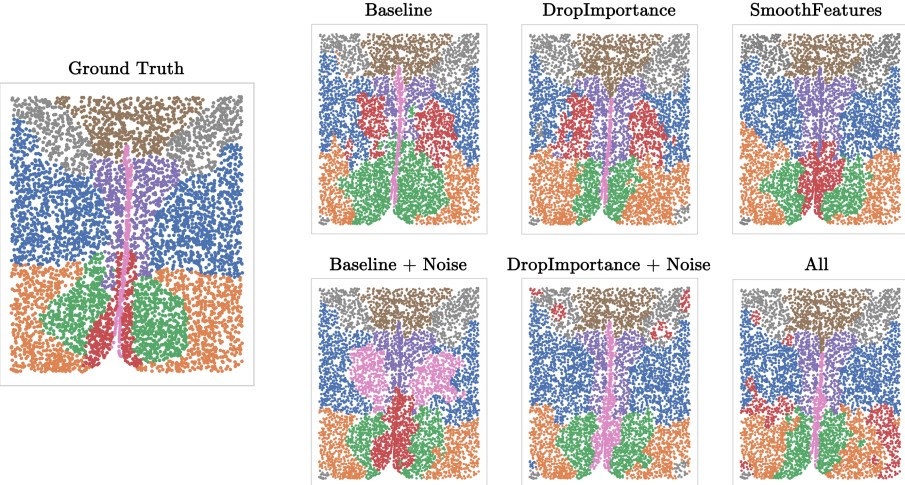

Figure 4: **Predicted and ground-truth domains in MERFISH tissue.** Visualization of a representative mouse brain sample. The left-most panel shows expert-annotated ground truth; remaining panels display predicted domains under different augmentation strategies. Augmentations strongly influence segmentation quality, broadly consistent with the quantitative results.

## 3.2   Supervised phenotype prediction in human cancer tissue

We evaluated augmentation strategies on relapse prediction in NSCLC samples, with performance measured by F1 score and AUROC (Tables 4 and 5). Models were trained with baseline augmentations (*DropFeatures* and *DropEdges*) and compared against advanced augmentations. In *PhenotypeShift*, we incorporated biologically motivated cell state transitions, including tumor adaptation to hypoxia, fibroblast subtype plasticity, and T cell differentiation into regulatory, exhausted, or proliferative states, while also accounting for myeloid–neutrophil annotation noise.

Under BGRL, the baseline achieved F1 = 0.59 and AUROC = 0.60. *FeatureNoise* improved AUROC to 0.61, while *DropImportance* alone decreased performance. The best F1 was obtained with *PhenotypeShift* (0.64), while the best AUROC was achieved by *FeatureNoise* (0.61). Combining all augmentations yielded intermediate gains (F1 = 0.62, AUROC = 0.60).

Under GRACE, the results showed similar patterns. The best F1 score was obtained with *DropImportance + FeatureNoise* (0.63), while the best AUROC was achieved with *Baseline + FeatureNoise* (0.59). The second best scores were achieved using *PhenotypeShift* both in terms of F1 score (0.63) and AUROC (0.59). Adding all augmentations together did not improve over single strategies. Overall, these results indicate that for phenotype prediction in cancer, both noise-based and biologically motivated augmentations may improve performance, but combining them provides only limited additional benefit, if at all.

## 3.3   Representation Analysis

To understand how augmentations influence the learned embedding geometry, we compared encoder representations across all configurations using linear CKA. Results of this representation analysis are presented in Figure 5.

Table 4: **Performance on phenotype prediction task using BGRL.** Relapse prediction in NSCLC samples using BGRL pretraining with different augmentation strategies. Reported as mean ± standard deviation across 5 random seeds. The best and second-best results by mean are highlighted.

| Augmentations | F1 Score | AUROC |
|---|---|---|
| Baseline | $0.5896 \pm 0.0213$ | $0.5986 \pm 0.0142$ |
| Baseline + FeatureNoise | $0.6265 \pm 0.0155$ | $\mathbf{0.6084 \pm 0.0097}$ |
| DropImportance | $0.6171 \pm 0.0245$ | $0.5848 \pm 0.0031$ |
| DropImportance + FeatureNoise | $0.6277 \pm 0.0011$ | $0.5665 \pm 0.0106$ |
| PhenotypeShift | $\mathbf{0.6375 \pm 0.0090}$ | $0.6006 \pm 0.0098$ |
| DropImp. + FeatNoise + PhenotypeShift | $0.6218 \pm 0.0291$ | $0.6030 \pm 0.0100$ |

Table 5: **Performance on phenotype prediction task using GRACE.** Relapse prediction in NSCLC samples using GRACE pretraining with different augmentation strategies. Reported as mean ± standard deviation across 5 random seeds. The best and second-best results by mean are highlighted.

| Augmentations | F1 Score | AUROC |
|---|---|---|
| Baseline | $0.6157 \pm 0.0125$ | $0.5759 \pm 0.0194$ |
| Baseline + FeatureNoise | $0.6208 \pm 0.0142$ | $\mathbf{0.5932 \pm 0.0090}$ |
| DropImportance | $0.6137 \pm 0.0355$ | $0.5707 \pm 0.0074$ |
| DropImportance + FeatureNoise | $\mathbf{0.6338 \pm 0.0052}$ | $0.5545 \pm 0.0122$ |
| PhenotypeShift | $0.6318 \pm 0.0096$ | $0.5897 \pm 0.0167$ |
| DropImp. + FeatNoise + PhenotypeShift | $0.6070 \pm 0.0409$ | $0.5870 \pm 0.0088$ |

For the domain identification task, most single augmentations yielded representations that were moderately similar to the baseline model. FeatureNoise, DropImportance, and SmoothFeatures all produced CKA values between 0.72 and 0.88, indicating that these transformations preserve the global structure of the tissue manifold while inducing controlled shifts in representation geometry. DropImportance produced a noticeable but not drastic shift from the baseline (CKA 0.76). In contrast, combining DropImportance with FeatureNoise caused a substantial reduction in similarity (CKA 0.52 to 0.60 relative to all the other models). This indicates that the combined perturbations interfere with the invariances induced by DropImportance and disrupt the structure of the embedding.

The phenotype prediction task showed a markedly different pattern. Here, only PhenotypeShift maintained moderate similarity to the baseline (CKA about 0.75). All other augmentations produced very low similarity to baseline embeddings (CKA below 0.2). This suggests that variation here is more easily overwritten by overly strong or misaligned invariances, leading to large shifts in the representation space. This pattern is consistent with PhenotypeShift being the most effective augmentation for this task.

These findings show that the geometry of the learned representation reacts differently to augmentation choice depending on the tissue context. Domain identification in healthy tissue benefits from controlled geometric shifts that emphasize structural regularities in tissue, whereas phenotype prediction in cancer tissue is sensitive to augmentations that distort the underlying topological structure.

## 4 Discussion

We systematically evaluated the role of graph augmentations in self-supervised GNN pretraining for spatial omics, using both BGRL [17] and GRACE [16] across two tasks: domain identification and phenotype prediction. While the absolute performance scores are modest, this primarily reflects the intrinsic difficulty and noise of these tasks in spatial omics data. Our models nonetheless reach performance levels comparable to recent state-of-the-art approaches, demonstrating the validity of the setup. The results show that augmentation choice has a decisive impact on downstream performance. In line with prior contrastive learning work [13–15], we find that well-aligned augmentations can enhance representations by encoding task-relevant invariances, whereas overly strong or misaligned transformations can degrade performance.

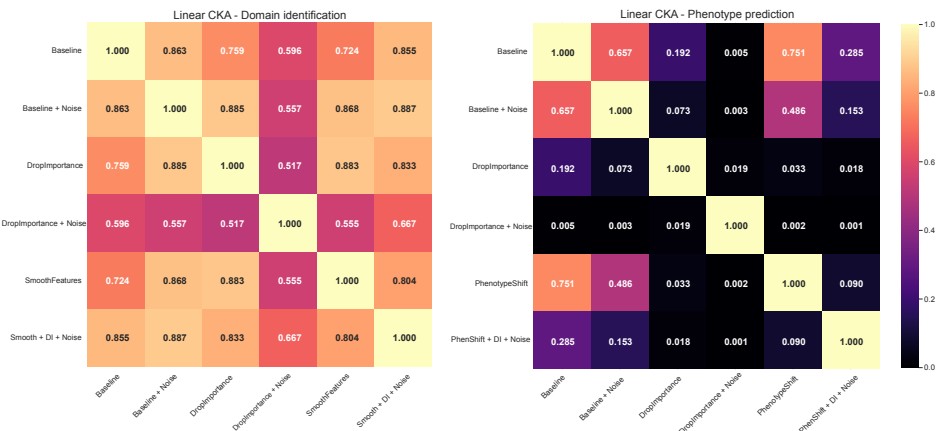

Figure 5: **Representational analysis using CKA.** Linear CKA similarity matrices comparing encoder representations across all augmentation configurations in the domain and phenotype tasks. The values quantify how similarly each configuration structures the embedding space and highlight augmentation induced shifts in representation geometry.

Domain identification benefited most from structural perturbations, with *DropImportance* improving performance by removing structurally redundant nodes and edges, albeit at the cost of additional computational overhead due to the need to compute and rank node and edge importance scores. In contrast, phenotype prediction showed limited gains from structural perturbations and instead improved with noise-based and biologically motivated augmentations such as *FeatureNoise* and *PhenotypeShift*. Composing these augmentations, however, provided only limited additional benefit or even hurt performance. A likely explanation is that the combined perturbations either dilute informative signal or exceed the capacity of the model to leverage additional invariances in this noisy, small-sample setting. These divergent outcomes highlight the distinct demands of the two tasks: in healthy tissue, domain identification profits from invariance to redundant structure, as tissue compartments are relatively stable, whereas in heterogeneous cancer tissue, phenotype prediction must contend with biological variability and label uncertainty [22–24].

Our findings parallel those in image and graph contrastive learning. SimCLR [13] demonstrated that augmentation design largely determines representation quality, while BYOL [14] and BGRL [17] highlighted that not all perturbations are beneficial. GRACE [16] also emphasized the role of augmentation strategies for graphs. Our results extend these insights to spatial omics: invariances induced by augmentations must be carefully matched to biological and experimental noise characteristics to improve downstream generalization. This has direct implications for the use of spatial omics in translational research, where robust embeddings can support diagnostics and patient stratification.

This study was limited to two downstream tasks and a curated set of augmentations. Domain identification was based on a small number of annotated healthy tissue samples, constraining statistical power, while phenotype prediction was restricted to a single cancer cohort. Moreover, the two tasks differed both in supervision regime and biological complexity, making it difficult to disentangle whether augmentation effectiveness depends primarily on task type (unsupervised vs. supervised) or tissue context (healthy vs. cancerous). Future work could address this by including supervised tasks on healthy tissue and unsupervised tasks on cancer tissue. Beyond these design choices, broader evaluation on additional omics technologies and clinical endpoints will be important. More extensive augmentation design could directly support translational use cases such as patient stratification or treatment response prediction, where robustness to biological noise is critical.

In summary, augmentation design is a critical factor in self-supervised learning on spatial omics graphs. Effective augmentations encode biologically plausible invariances, improving model robustness and downstream accuracy, while misaligned ones can add cost without benefit. Our results reinforce the view that augmentation choice is not incidental but a central design decision in graph contrastive learning.

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
