# A  Appendix / supplemental material

## A.1  Bootstrapped Graph Latents (BGRL)

Bootstrapped Graph Latents (BGRL) [1] is a self-supervised graph representation learning method used in this project. It avoids labels and negative samples by predicting alternate augmentations of the same input graph.

A graph $\mathbf{G} = (\mathbf{X}, \mathbf{A})$ is first augmented into two alternate views $\mathbf{G}_1 = (\tilde{\mathbf{X}}_1, \tilde{\mathbf{A}}_1)$ and $\mathbf{G}_2 = (\tilde{\mathbf{X}}_2, \tilde{\mathbf{A}}_2)$ via graph augmentation functions $\mathcal{T}_1$ and $\mathcal{T}_2$, respectively. An online encoder $\mathcal{E}_\theta$ with parameters $\theta$ then produces an online representation from the first augmented view, $\tilde{\mathbf{H}}_1 := \mathcal{E}_\theta(\tilde{\mathbf{X}}_1, \tilde{\mathbf{A}}_1)$, and a target encoder $\mathcal{E}_\phi$ with parameters $\phi$ produces a target representation from the second augmented view, $\tilde{\mathbf{H}}_2 := \mathcal{E}_\phi(\tilde{\mathbf{X}}_2, \tilde{\mathbf{A}}_2)$. A prediction of the target representation, $\tilde{\mathbf{Z}}_1 := p_\theta(\tilde{\mathbf{H}}_1)$, is obtained by feeding the online representation into a node-level predictor $p_\theta$.

To update the online encoder's parameters $\theta$, the gradient of the cosine similarity of the predicted target representation $\tilde{\mathbf{Z}}_1$ and the true target representation $\tilde{\mathbf{H}}_2$ is computed with respect to $\theta$:

$$l(\theta, \phi) = -\frac{2}{N} \sum_{i=0}^{N-1} \frac{\tilde{\mathbf{Z}}_{(1,i)} \tilde{\mathbf{H}}_{(2,i)}^\top}{\left\| \tilde{\mathbf{Z}}_{(1,i)} \right\| \left\| \tilde{\mathbf{H}}_{(2,i)} \right\|} \tag{1}$$

$$\theta \leftarrow \mathrm{optimize}(\theta, \eta, \partial_\theta l(\theta, \phi)). \tag{2}$$

Here, $\eta$ is the learning rate and in practice, the loss is symmetrized by also predicting the target representation of the first view with the online representation of the second view.

The target encoder's parameters $\phi$ are updated as an exponentially moving average with decay rate $\tau$ of the online encoder's parameters $\theta$:

$$\phi \leftarrow \tau\phi + (1 - \tau)\theta. \tag{3}$$

## A.2  Deep Graph Contrastive Representation Learning (GRACE)

Deep Graph Contrastive Representation Learning (GRACE) [2] is a self-supervised method for unsupervised graph representation learning. Unlike methods relying on global readouts, GRACE directly contrasts node-level embeddings across two randomly corrupted views of the same graph.

Formally, given a graph $\mathbf{G} = (\mathbf{X}, \mathbf{A})$, GRACE generates two augmented views $\mathbf{G}_1 = (\tilde{\mathbf{X}}_1, \tilde{\mathbf{A}}_1)$ and $\mathbf{G}_2 = (\tilde{\mathbf{X}}_2, \tilde{\mathbf{A}}_2)$ by applying stochastic corruption functions $\mathcal{T}_1, \mathcal{T}_2$ to features and edges. Specifically, GRACE uses (i) *edge removal* with probability $p_r$ and (ii) *feature masking* with probability $p_m$ to generate diverse contexts.

A shared GNN encoder $f_\theta$ then computes node embeddings $\mathbf{U} = f_\theta(\tilde{\mathbf{X}}_1, \tilde{\mathbf{A}}_1)$ and $\mathbf{V} = f_\theta(\tilde{\mathbf{X}}_2, \tilde{\mathbf{A}}_2)$. For a node $i$, the embeddings $(\mathbf{u}_i, \mathbf{v}_i)$ from the two views form a positive pair, while embeddings from other nodes act as negatives. The similarity between two embeddings is estimated by a critic

$$\theta(\mathbf{u}, \mathbf{v}) = \frac{g(\mathbf{u})^\top g(\mathbf{v})}{\|g(\mathbf{u})\| \, \|g(\mathbf{v})\|}, \tag{4}$$

where $g(\cdot)$ is a two-layer projection head and the similarity is scaled by a temperature $\tau$.

The contrastive loss for node $i$ is defined as

$$\ell(\mathbf{u}_i, \mathbf{v}_i) = -\log \frac{\exp(\theta(\mathbf{u}_i, \mathbf{v}_i)/\tau)}{\exp(\theta(\mathbf{u}_i, \mathbf{v}_i)/\tau) + \sum_{k \neq i} \exp(\theta(\mathbf{u}_i, \mathbf{v}_k)/\tau) + \sum_{k \neq i} \exp(\theta(\mathbf{u}_i, \mathbf{u}_k)/\tau)}. \tag{5}$$

The final symmetric objective averages over all nodes:

$$J = \frac{1}{2N} \sum_{i=1}^{N} \left[ \ell(\mathbf{u}_i, \mathbf{v}_i) + \ell(\mathbf{v}_i, \mathbf{u}_i) \right]. \tag{6}$$

## A.3 Augmentation benchmark

To assess the computational costs associated with different augmentations and combinations of augmentations, they were applied to synthetic graphs of varying sizes while measuring runtime and memory usage.

For augmentations relevant to domain identification, synthetic graphs were generated to mimic the structure of real domain identification data. These graphs consisted of nodes with 50 numerical features, with feature similarities reflecting group structures, i.e., nodes within a group had more similar features than those in different groups. For phenotype prediction augmentations, graphs were designed to contain nodes annotated with a cell type feature and a cell size feature. Additionally, edges were annotated with a binary indicator distinguishing "near" from "distant" connections.

All individual augmentations applicable to either domain identification or phenotype prediction were tested on their respective synthetic graph types. Furthermore, combinations of augmentations, corresponding to those evaluated in the main experiments, were also benchmarked. Each augmentation or combination was applied to synthetic graphs of increasing size, with each experiment repeated three times on a single GPU. For each run, both the runtime and peak GPU memory usage were recorded. The mean values across the three replicates were reported as the final result.

The results for domain identification augmentations are shown in Figure 1. Augmentation modes using *DropImportance* exhibit higher runtime compared to baseline augmentations (*DropFeatures* and *DropEdges*) and noise-based augmentations (*SpatialNoise* and *FeatureNoise*), though still running for 1 second or less for all graph sizes. Smoothing exhibits the highest memory usage of all the augmentations.
**Note:** The relatively high runtime observed for smaller graphs primarily reflects fixed computational overheads (e.g., data loading, graph construction, and GPU initialization), which dominate when per-graph computation is fast. These effects diminish as graph size increases, where runtime scales more proportionally with the number of nodes and edges.

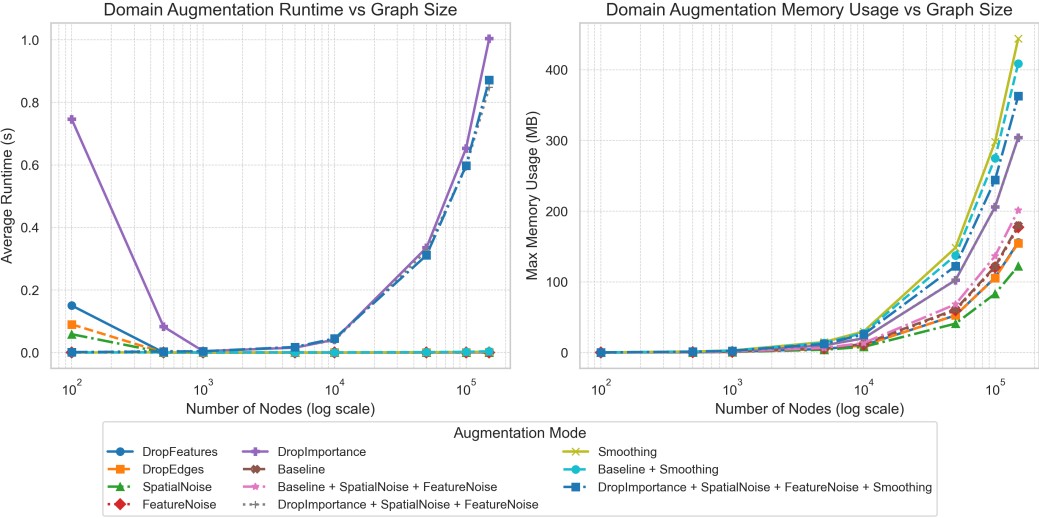

Figure 1: **Benchmark of domain identification augmentations.** Runtime (left) and peak GPU memory usage (right) for domain identification augmentations across increasing graph sizes. Each line represents either an individual augmentation or a combination of augmentations.

The results for phenotype prediction augmentations are shown in Figure 2. The runtime scaling trends are similar to those in the domain identification results. Augmentation modes using *DropImportance* scale worse than baseline and noise-based augmentations in both runtime and memory usage.

Overall, the benchmark highlights substantial variability in the computational efficiency of different augmentation strategies. Especially more complex augmentations, such as *DropImportance* and

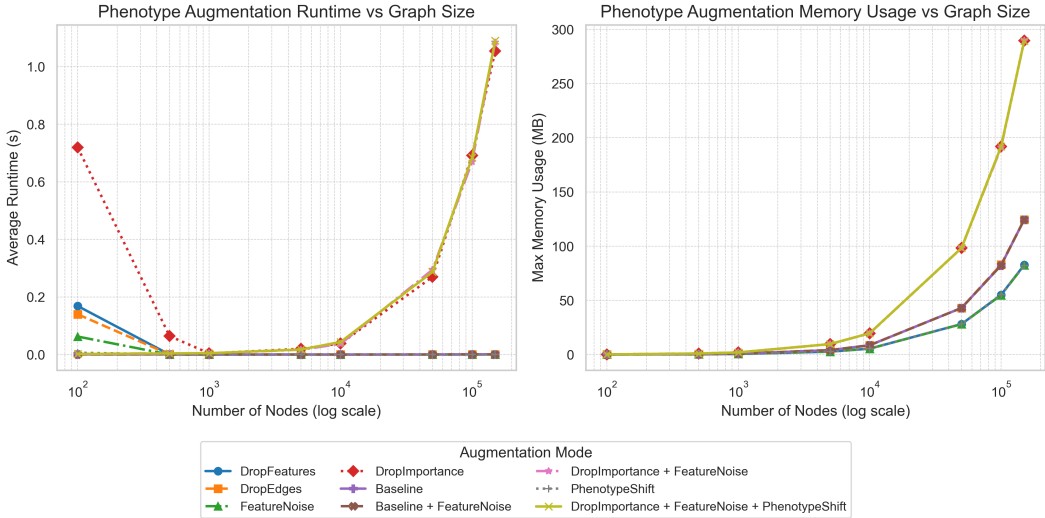

Figure 2: **Benchmark of phenotype prediction augmentations.** Runtime (left) and peak GPU memory usage (right) for phenotype prediction augmentations across increasing graph sizes. Each line represents either an individual augmentation or a combination of augmentations.

*Smoothing*, significantly increase runtime and memory consumption on large graphs, which also introduces considerable computational overhead during model training.

### A.4 Classification metrics

To assess the performance of the phenotype prediction model, several binary classification metrics were used. These were computed from the predicted logits $\mathbf{z} \in \mathbb{R}^N$ and the ground truth binary labels $\mathbf{y} \in \{0, 1\}^N$ for all $N$ samples.

First, the predicted logits were transformed into probabilities using the sigmoid function:

$$\hat{\mathbf{p}} = \sigma(\mathbf{z}) = \frac{1}{1 + e^{-\mathbf{z}}} \tag{7}$$

A threshold $\tau \in [0, 1]$ was applied to convert probabilities into binary predictions:

$$\hat{\mathbf{y}} = \mathbb{I}[\hat{\mathbf{p}} \geq \tau] \tag{8}$$

During validation, the threshold $\tau$ was chosen to maximize the F1 score across a set of candidate thresholds. Once the optimal threshold was selected, the following metrics were computed:

- **AUROC (Area Under the Receiver Operating Characteristic Curve)**: The AUROC quantifies the probability that a randomly chosen positive sample is ranked higher than a randomly chosen negative sample by the model's scoring function. Formally, if $s(x)$ denotes the prediction score, then

$$\text{AUROC} = \mathbb{P}\big(s(x^+) > s(x^-)\big),$$

where $x^+$ and $x^-$ are independent draws from the positive and negative classes, respectively.

Equivalently, AUROC corresponds to the area under the curve tracing the true positive rate (TPR) against the false positive rate (FPR) as the classification threshold is varied:

$$\text{TPR}(t) = \frac{\text{TP}(t)}{\text{TP}(t) + \text{FN}(t)}, \quad \text{FPR}(t) = \frac{\text{FP}(t)}{\text{FP}(t) + \text{TN}(t)},$$

where TP, FP, TN, FN denote true/false positives/negatives at threshold $t$. A value of $0.5$ corresponds to random guessing, while $1.0$ indicates perfect class separability.

- **Precision**: Fraction of predicted positives that are correct:

$$\text{Precision} = \frac{TP}{TP + FP} \tag{9}$$

- **Recall (Sensitivity)**: Fraction of actual positives that are correctly identified:

$$\text{Recall} = \frac{TP}{TP + FN} \tag{10}$$

- **F1 Score**: Harmonic mean of precision and recall, balancing both metrics:

$$\text{F1} = 2 \cdot \frac{\text{Precision} \cdot \text{Recall}}{\text{Precision} + \text{Recall}} \tag{11}$$

## A.5 Clustering evaluation metrics

To evaluate the quality of clustering results obtained, three metrics were employed: Normalized Mutual Information (NMI), Homogeneity, and Completeness. These metrics assess how well the predicted clustering aligns with ground truth domain labels.

*NMI* measures the mutual dependence between the predicted clustering $C$ and the ground truth labels $Y$, normalized by the entropy of both. It is defined as:

$$\text{NMI}(C, Y) = \frac{2 \cdot I(C; Y)}{H(C) + H(Y)} \tag{12}$$

where $I(C; Y)$ is the mutual information between $C$ and $Y$, and $H(\cdot)$ denotes entropy. Mutual information is given by:

$$I(C; Y) = \sum_{c \in C} \sum_{y \in Y} P(c, y) \log \left( \frac{P(c, y)}{P(c)P(y)} \right) \tag{13}$$

Here, $P(c, y)$ is the joint probability of a sample being in cluster $c$ and class $y$, while $P(c)$ and $P(y)$ are the marginal probabilities.

*Homogeneity* assesses whether each cluster contains only data points that belong to a single class. It is defined as:

$$\text{HOM}(C, Y) = \begin{cases} 1 & \text{if } H(Y|C) = 0 \\ 1 - \frac{H(Y|C)}{H(Y)} & \text{otherwise} \end{cases} \tag{14}$$

where $H(Y|C)$ is the conditional entropy of the ground truth labels given the cluster assignments, and $H(Y)$ is the entropy of the ground truth.

*Completeness* measures whether all members of a given class are assigned to the same cluster. It is defined as:

$$\text{COM}(C, Y) = \begin{cases} 1 & \text{if } H(C|Y) = 0 \\ 1 - \frac{H(C|Y)}{H(C)} & \text{otherwise} \end{cases} \tag{15}$$

where $H(C|Y)$ is the conditional entropy of the predicted cluster assignments given the true class labels.

## A.6 Possible cell type transitions for the *PhenotypeShift* augmentation

We allow a restricted set of biologically motivated cell type transitions, reflecting known plasticity and differentiation processes in the tumor microenvironment:

- **Tumor adaptation:** Tumor cells (normal) can transition to hypoxic tumor states [3].
- **Fibroblast (CAF) plasticity:** Collagen CAFs may become myofibroblastic CAFs (mCAFs) or adapt to hypoxia; mCAFs can further switch into SMA$^+$ CAFs, PDPN$^+$ CAFs, vascular CAFs, or hypoxic CAFs; iCAFs can adopt PDPN$^+$ or IDO$^+$ states; IDO$^+$ CAFs can also adapt to hypoxia; tumor-promoting CAFs (tCAFs) can transition to hypoxic tCAFs [4–6].

- **CD4$^+$ T cell differentiation:** CD4 T cells can give rise to regulatory T cells (Tregs), PD1$^+$ exhausted cells, IDO$^+$ subsets, proliferative (Ki67$^+$) states, or TCF1/7$^+$ progenitor-like cells [7, 8].

- **CD8$^+$ T cell differentiation:** CD8 T cells can give rise to IDO$^+$ subsets, proliferative (Ki67$^+$) states, or TCF1/7$^+$ progenitor exhausted cells [8, 9].

- **Myeloid refinement:** Myeloid cells can be further refined into neutrophil identities, reflecting annotation resolution rather than a true biological transition [10].

## A.7 Hyperparameter ranges used for tuning augmentations

Table 1: **Hyperparameter search ranges for graph augmentations.** For each augmentation, the tuned hyperparameters and their respective ranges are listed. Intervals denote uniform sampling from the specified range.

| Augmentation | Hyperparameter | Range |
|---|---|---|
| DropEdges | $p$ | [0.1, 0.4] |
| DropFeatures | $p$ | [0.1, 0.4] |
| DropImportance | $\lambda_p$ $\mu$ | [0.4, 0.6] [0.1, 0.4] |
| SpatialNoise | $\sigma_{\text{spatial}}$ | [2.0, 30.0] |
| FeatureNoise | $\sigma_{\text{feature}}$ | [0.05, 1.0] |
| SmoothFeatures | $\alpha$ | [0.0, 0.5] |
| PhenotypeShift | $p_{\text{shift}}$ | [0.0, 0.3] |