# OpenReview forum: "Exploring Augmentation-Driven Invariances for Graph Self-supervised Learning in Spatial Omics"
_NeurIPS.cc/2025/Workshop/UniReps — UniReps2025_

### Official Review · Reviewer_E4k2 · 2025-09-12
**How does different augmentations during SSL training affect downstream tasks**

**Confidence:** 4

**Review:**

### Summary:

This paper explores how different augmentations used during graph self-supervised learning affect two downstream tasks: clustering and prediction.

### Pros:
- The paper is overall well-written. The presentation is clear, the topic is well motivated, and the writing is neatly organized.
- The experiments on the considered augmentations are relatively comprehensive.
- Limitations are acknowledged.

### Cons:
- Although there are two baseline augmentations, I wonder how do the downstream models perform without any augmentation.
- The authors mentioned that there are some augmentations that exist in the literature and some are novel, but it's unclear in Section 2.2 which ones are novel and which ones are not.
- The augmentation methods included are rather generic, and it is unclear to me what does "well-aligned" mean. The authors mentioned if the augmentations are "well-aligned" then its helpful in the abstract and the discussion, but this term seems not really discussed in the main body of the paper. How do the authors classify what is "well-aligned" and align with what exactly?
- The paper explores some augmentation methods and their combinations, but in terms of the strategy of augmentations, as mentioned in the abstract is still a bit unclear.

**Score:**

4

**Topic Fit:**

2

---

### Official Review · Reviewer_uxFL · 2025-09-13
**Well executed biological experiments, deeper mechanistic analyses needed.**

**Confidence:** 3

**Review:**

The authors propose a systematic audit of graph embedding augmentations, across self-supervised methods, for spatial omics data. Biologically motivated graph augmentations are proposed as a framework for evaluation. Two central tasks in biology are assessed:  unsupervised domain identification and supervised phenotype prediction, using the BGRL and GRACE frameworks.

**Strengths**:

**Elegant framework:** atomic graph actions are well defined in a clear taxonomy and have diversity in both breadth and levels of complexity (baselines vs advanced).

**Thorough evaluation:** Computational benchmarking, statistical testing across runs.

**Relevant problems** Important challenges in the single cell community are addressed through this work.

**Weaknesses**:

**Limited experimental scope:** while breadth is appreciated, a single task per supervision type  is insufficient to establish general augmentation principles. In addition, there’s 12 total samples in domain identification, which may limit statistical power. More fundamentally, the authors treat invariances as a design goal, something to be learned, but there’s limited justification as to why we’d like to keep biological invariances. Do the learned invariances correspond to biological mechanisms? Tissue regions? Why are they interesting for the proposed application but more importantly, for a method to be able to include this information within convergent representations?

**Limited Novelty:** DropImportance seems to be the consistent top performer as an augmentation, by a considerable margin, but it is essentially importance-weighted dropout. PhenotypeShift seems to outperform all methods in the phenotype prediction task in isolation. However, combining top-performing augmentations yields worse results than individual methods, with no explanation as to why three augmentations together underperform a gaussian noise baseline. More clarity is required on augmentation interactions.
Missing a focus on representations: Beyond qualitative visualizations, the paper lacks representational analysis entirely. There’s no investigation of whether BGRL and GRACE learn similar embeddings under identical augmentations. Essentially: how are the learned representations across invariances the same or different? It is difficult to assess if the methods capture biological mechanisms or if they’re enforcing similar graph-based artifacts on downstream performance metrics alone.

**Insufficient mechanistic explanations:** The authors claim DropImportance works by "removing structurally redundant nodes and edges," but their results contradict this theory. If redundancy removal were the mechanism, wouldn’t adding FeatureNoise enhance performance by forcing focus on truly important structural features? Instead, DropImportance alone (0.6585 NMI) outperforms DropImportance + FeatureNoise (0.6488 NMI), which could result from FeatureNoise masking the importance signals that DropImportance relies on. These results, as presented, don't show a clear explanation on how these augmentations interact, which undermines claims of principled augmentation design. More broadly, they offer biological interpretations ("tissue compartments are stable") without demonstrating that their augmentations specifically target these claimed mechanisms.

**Recommendation**
While the work is presented in a scientifically sound way and the biological experiments are well executed, results require a substantial revision to fit the scope for this workshop. Representational analyses beyond loosely defined clusters, characterizing the learned invariances per method and perhaps a deeper mechanistic investigation of why or when self-supervised methods converge, given the augmentations, would all be in order. Recommending rejection in its current form.

**Score:**

2

**Topic Fit:**

1

---

### Official Review · Reviewer_Phz1 · 2025-09-13

**Confidence:** 3

**Review:**

This paper demonstrates how different graph augmentation strategies influence node and graph embeddings in spatial omics data. It evaluates both established methods and new biologically motivated augmentations across two tasks: domain identification in healthy tissue and phenotype prediction in cancer. Through extensive numerical study, I believe the findings demonstrate how inductive biases and invariances impact downstream performance, introducing novel augmentations tailored to spatial omics.

**Score:**

4

**Topic Fit:**

3

---

### Official Review · Reviewer_oYfF · 2025-09-16
**An analysis of pre-training augmentations for GIN on -omics data**

**Confidence:** 4

**Review:**

## Summary

This work explores self-supervised graph augmentation and training objectives for spatial-omics data, an emerging resource of expression data (e.g., RNA or Protein) which preserves spatial context within cellular or biological processes. The authors gather a number of graph augmentation techniques, and evaluate them on domain identification and phenotype prediction. Several “basic” (Drop Features, Drop Edges) and “advanced” (Drop Importance, Spatial Noise, Feature Noise, etc) are evaluated. Augmentation techniques are evaluated on 3 publically available datasets.

## Strengths

- The main strength of this evaluation I believe is timeliness, as spatial -omics analysis will without doubt play a large role in our future understanding of biological mechanisms and disease
- The authors conduct a thorough analysis and also show that too many augmentation strategies adversely affect performance. This is a useful guide for practitioners and could be considered as a “best practices” paper.

## Weaknesses

- Besides providing qualitative and quantitative analysis of augmentation techniques in a very specific domain, the work does not bring much else to the table. The work lacks novelty.
- I worry that, as a result, this would not interest a wider audience at NeurIPS, and may be more suitable for a domain-specific venue targeting practitioners.
- The manuscript could be improved by looking at other models besides GIN, which is certainly also helpful for practitioners. The resulting patterns that emerge across networks could be analyzed.

**Score:**

1

**Topic Fit:**

1